# Distance from the Community to the City Center Is a Vital Determinant of Dietary Diversity Score for Rural Community-Dwelling Older Adults in Taiwan

**DOI:** 10.3390/nu17081318

**Published:** 2025-04-10

**Authors:** Chiu-Ying Chen, Hung-Kuan Yen, I-Hui Wu, Yi-Chien Lu, Ning-Huei Sie, Shau-Huai Fu, Chen-Yu Wang

**Affiliations:** 1Department of Public Health, College of Medicine, National Cheng Kung University, Tainan 70101, Taiwan; wggwze@gmail.com; 2Department of Administration, Tainan Hospital, Ministry of Health and Welfare, Tainan 70043, Taiwan; 3Department of Orthopedics, National Taiwan University Hospital, Taipei 10051, Taiwan; eric.yen.a92@gmail.com; 4Department of Geography, National Taiwan University, Taipei 10617, Taiwan; flowerfrog@gmail.com; 5National Center for Geriatrics and Welfare Research, National Health Research Institutes, Huwei 632007, Taiwan; carrielu2011@gmail.com (Y.-C.L.); eaglea0531@gmail.com (N.-H.S.); 6Center of General Education, National Chung Cheng University, Chiayi 62102, Taiwan; 7Department of Orthopedics, National Taiwan University Hospital Yun-Lin Branch, Douliu 640203, Taiwan; 8Department of Pharmacy, National Taiwan University Hospital Yun-Lin Branch, Douliu 640203, Taiwan

**Keywords:** distance, dietary diversity score, elderly, rural communities, malnutrition

## Abstract

**Background:** There was little knowledge of the dietary diversity of older adults in rural areas, with limited studies examining their nutritional status and related factors. This study aimed to assess the nutritional status of older adults in rural communities in Taiwan and explore the association between characteristics and the dietary diversity score (DDS). **Methods:** We collected data on sociodemographic, nutritional status, and DDS. The DDS was estimated based on nutritional intake using a 24 h dietary recall (24HR). Participants were grouped by distance from the city center into three categories (0–4 km, 4–8 km, and >8 km) to assess their correlation with food diversity. **Results:** A total of 567 participants were included, of whom 81.3% were women, and 65.5% were over 75 years of age. Of the participants, 154 lived within 4 km of the city center, 255 lived 4–8 km away, and 158 resided in remote communities (>8 km from the city center). Out of 538 participants who completed the dietary questionnaire, 274 (50.9%) were categorized as having low dietary diversity (DDS ≤ 4), while 264 (49.1%) were categorized as having high dietary diversity (DDS > 4). **Conclusions:** Our results showed the distance from the community to the city center significantly influences the DDS of the residents in rural communities. Nearly half of the rural population has a low DDS. The implementation of the research project aimed to determine the feasibility of daily nutrition evaluation and optimization programs for the elderly in rural areas is crucial in the future.

## 1. Introduction

Malnutrition and undernourishment are significant health problems among older adults [1]. Studies have shown that poor nutrition in this population is linked to decreased immune function, increased medical expenditure, and prolonged hospitalization in the UK and US [2]. Additionally, the inefficiency of ingested nutrition directly and indirectly increases the risk of death [3]. In older adults, optimal nutritional status is important not only for the prevention and treatment of various diseases [4] but also for promoting functional independence, thereby improving quality of life and promoting healthy aging [5].

There is a significantly higher proportion of older adults in rural areas compared to urban areas, and they face difficulties in purchasing food due to sparse populations and limited transportation options in Japan [6,7]. The consequences of urbanization resulting from social environmental developments and the displacement of young people have affected the physical and mental health of the elderly who remain in rural communities [8,9]. This has led to social isolation, loneliness, and a reduced food supply, which affects the quality and intake of food for older adults and increases the risk of malnutrition [10]. Previous studies in both Asia and Europe, have shown that loneliness affects the appetite of older adults, resulting in reduced regular meals, increased consumption of convenience foods, reduced quantities and types of food, and a further reduction in dietary intake that increases the risk of malnutrition [11,12].

The dietary diversity score (DDS) is a simple, fast evaluation tool that requires no equipment or complex measurement [13,14]. It quantifies the number of different foods or groups of foods consumed during a given period, serving as an important indicator of nutritional balance [15]. Previous studies in Asia found that older adults with higher dietary diversity tend to have better nutrient intake, improved energy levels, and lower mortality rates [13,16,17,18,19].

In the past, there was little knowledge of the dietary diversity of older adults in rural areas, with limited studies examining their nutritional status and related factors. This study, therefore, aimed to collect information on dietary diversity, nutritional status, and related factors among community-dwelling older adults in Yunlin, a rural area in Taiwan, and to provide health strategies for maintaining a healthy lifestyle for rural residents in the future.

## 2. Materials and Methods

### 2.1. Design and Setting

This study was a secondary, cross-sectional analysis of a community-based randomized controlled trial to evaluate the effectiveness of a multicomponent intervention on the intrinsic capacity, disability, and quality of life of older adults in rural communities.

The Healthy Longevity and Aging in Place (HOPE) study [20] is a four-year, open-label, stepped-wedge, three-armed cluster randomized controlled trial taking place in a rural community. This study commenced in November 2021 and will continue until the end of 2025. The HOPE study provides community-based integrated care and aims to enhance the quality-adjusted life years and prevent disability. Our team comprised orthopedic and rehabilitation doctors, pharmacists, physical therapists, dietitians, research nurses, and other professionals, delivering integrated, multifaceted healthcare services, addressing issues including osteoporosis, sarcopenia, polypharmacy, and providing support for exercise and nutrition to enhance the overall well-being of rural residents, both physically and mentally. In this study, we present the baseline nutritional status of this study population and explore the association between the baseline characteristics and the DDS.

This trial was approved by the Institutional Review Board of the National Taiwan University Hospital (protocol ID: 202106076RIND) and registered at ClinicalTrials.gov (NCT05104034). All the participants provided informed consent to participate in this study.

Recruitment and evaluation were performed at congregate meal services (CMS) centers in the communities of Yunlin County, a rural area in Taiwan with a high proportion of older adults and relatively lower average income levels. Approximately 150 CMS centers within Yunlin County serve nearly 3000 senior citizens daily. A range of community activities can be undertaken, and health services delivered to elderly individuals under the CMS infrastructure, providing a platform for social welfare activities and delivery of healthcare benefits.

### 2.2. Participants

We recruited 636 community residents from 30 CMS centers across rural, suburban, and urban areas of Yunlin County. All community-dwelling residents aged 50 years or older participating in the CMS were considered eligible. Exclusion criteria included terminally ill patients with a life expectancy of <3 months and individuals with severe cognitive impairments preventing communication. Finally, 567 participants who signed the consent form and completed the questionnaire were included.

### 2.3. Baseline Characteristics Collection

We interviewed each participant individually and collected data on gender, age, body composition (muscle mass), anthropometric measurements (height, weight, body mass index [BMI], calf circumference), residency status, education level, monthly income, and daily living activities, including the Basic Activities of Daily Living (BADL) questionnaire. A BADL score of 100 indicated independence, a score of 91–99 signified mild disability, and a score of 61–90 reflected moderate disability. Other assessments included 6-m gait speed, handgrip strength, frailty measured using the Study of Osteoporotic Fractures [21], and disease status assessed by the Charlson Comorbidity Index (CCI) [22]. To assess the potential impact of transportation barriers on food diversity in remote areas, we also recorded the distance from each CMS to the nearest city center as a proxy for food accessibility. Since Yunlin County is vast and elongated, we designated the centers of its two largest towns—Douliu Railway Station and Huwei Town Farmers Association—as reference city centers to calculate. The driving distances to neighboring CMS were categorized into three groups: 0–4 km, 4–8 km, and >8 km [23,24]. The 4 km cutoff was selected because it corresponds to the average daily walking distance for the elderly population, which is approximately 50 min per day [25] at an average walking speed of 80 m per minute. Distances within 4 km are typically accessible by walking. For distances greater than 4 km, alternative transportation, such as bicycles or motorcycles, may be required, with the 8 km cutoff representing the approximate distance that could be covered by bicycle in a 20-min ride. Distances exceeding 8 km are generally not easily accessible by bicycle or motorcycle for older adults. In these cases, reaching the city center would likely require public transportation or private vehicles, both of which may be challenging for elderly individuals, especially in rural areas with limited access to public transport. Among the 30 CMS centers, 70% are located more than 4 km, and 3% are more than 8 km from the city centers.

### 2.4. Nutritional Status

The Mini-Nutritional Assessment-Short Form (MNA-SF) is a valuable and efficient screening tool for identifying the risk of malnutrition in older adults [26,27,28]. It evaluates factors such as height, weight, appetite, weight loss, mobility, acute psychological stress, and neuropsychological problems [27]. MNA-SF scores ranged from 0 to 14, with a total of <8, 8–11, and >11 points representing malnourished, at risk of malnutrition, and normal nutritional status, respectively [29,30].

### 2.5. Dietary Intake Information and DDS

Dietary intake was assessed using 24-h dietary recall (24HR), a validated method known for its effectiveness, high responsiveness, and simplicity [31,32,33]. Standard food models were used to help individuals remember and record as accurately as possible the types and quantities of all liquids and solid foods eaten over the past 24 h [32,33,34]. In accordance with the recommendations developed by the Health Promotion Administration (HPA) under the Ministry of Health and Welfare (MOHW) [35], Taiwan, the dietary habits, deviations from the guidelines, and dietary balance of rural elderly people were assessed.

Dietary diversity was assessed using DDS [14], which estimates nutritional intake using 24HR and calculates DDS based on six major food groups [13]: dairy, soy/fish/eggs/meat, whole grains, fruit, vegetables, and oils and nuts. Each food group consumed by at least half the recommended daily portion was assigned a score of 1, with a maximum DDS of 6 points. DDS > 4 and ≤4 are considered as high and low dietary diversity, respectively [13].

### 2.6. Statistical Analysis

Descriptive statistics were presented as the mean ± standard deviations (SDs) and percentage (%) for continuous and categorical variables, respectively. Analysis of variance (ANOVA) and Chi-square tests were used to compare sequential and classification variables. Multivariable logistic regressions were used to evaluate the associations between participant characteristics and DDS, with associations presented as adjusted odds ratios (aORs) and 95% confidence intervals (CIs). The *p*-values were two-tailed, with *p* < 0.05 considered statistically significant. A stepwise selection was further adopted for the most prominent variables associated with DDS. All statistical analyses were conducted using SPSS version 17.0.

## 3. Results

### 3.1. Characteristics of the Participants

The descriptive characteristics of the participants are summarized in Table 1. Of the 567 participants, 81.3% were women, and 65.5% were over 75 years of age. The average BMI was 24.7 ± 3.7 kg/m^2^, with a prevalence of overweight and obesity at 53.9% (Appendix A). One-fifth of the population lived alone, with 154 living in communities within 4 km of the city center, 255 living 4–8 km away from the city center, and 158 residing in remote communities >8 km away from the city center. 67.5% had a monthly income below USD 300. A total of 77% of the population were illiterate and under-academic in the country. Overall, the average BADL score was 98.64 ± 4.85, with 3.3% and 7.1% experiencing mild and moderate disability, respectively.

Regarding nutritional status assessment, the MNA-SF identified 5.5% of participants as malnourished ratio. Of the 538 respondents who completed the 24HR questionnaire, the average dietary diversity score was 4.5 ± 0.8. The average calorie and protein intake was 1287 ± 395 kcal/day and 49.0 ± 19.2 g/day, with under-intake ratios of 44.3% and 43.2%, respectively (Appendix A). Out of 538 participants who completed the dietary questionnaire, 274 (50.9%) were categorized as having low dietary diversity (DDS ≤ 4), while 264 (49.1%) had high dietary diversity (DDS > 4). The under-consuming of dairy products, fruits, and vegetables was 71.1%, 51.6%, and 14.7%, respectively (Table 1).

### 3.2. Association Between Baseline Characteristics and Low or High DDS

Compared to those with low dietary diversity, participants with high dietary diversity were younger with higher educational levels, fewer people living >8 km from the city center, higher monthly incomes, lower rates of sarcopenia and frailty, faster 6 m walking speeds, greater handgrip strength, better BADL scores, higher total daily calorie intake, and higher intake of the three main nutrients (carbohydrate, fat, protein) (*p* < 0.05) (Table 2) in the ANOVA and Chi-square tests.

### 3.3. Associations Between the Characteristics and Risk of DDS

The multivariate logistic regression analysis revealed that factors such as sarcopenia, distance to the city center, and calorie and protein intake were significantly associated with DDS (Table 3). Stepwise selection further identified that distance to the city center, sarcopenia, and total daily calorie intake were key predictive factors. Figure 1 demonstrates the DDS score distribution and the distance from the community to the city center among the older adults of all communities included in this research.

## 4. Discussion

This study provides valuable insights into the nutritional status of older adults in rural communities and the factors affecting nutritional diversity. We found that approximately 50% of participants had low DDS, mainly due to inadequate consumption of dairy (71.1%), fruits (51.6%), and vegetables (14.7%). This suggests that the intake of calcium and dietary fiber in these individuals is likely insufficient. Additionally, we found that the distance from the city center was significantly associated with DDS, with an aOR of 2.17 (95% CI: 1.24–3.78, *p* = 0.006). This suggests that elderly individuals living in communities more than 8 km from the city center face more than twice the risk of having a lower DDS compared to those residing within 0–4 km. In epidemiological and public health contexts, an odds ratio (OR) greater than 2 is generally considered to represent a moderate to strong effect, with potential clinical significance (reference). This finding suggests that distance from the city center may serve as a predictor of DDS, which, in turn, could be linked to clinical outcomes such as increased frailty or malnutrition. In addition to this distance, sarcopenia and total daily calorie intake were identified as key predictive factors of DDS through stepwise selection.

Taiwanese dietary habits include the consumption of dairy products, which are also incorporated into the Taiwanese National Health Administration’s daily dietary guidelines [35] with recommended intake levels. Fresh milk is the primary dairy product consumed in Taiwan, and Yunlin County is one of the major dairy farming regions. In the past, Taiwan’s developing society faced challenging growth conditions, where dietary habits prioritized quantity (eating until full) over quality. Over time, this led to frugal eating habits among the elderly, resulting in reduced consumption. Additionally, Yunlin County is a major agricultural region in Taiwan, facing significant outmigration of its young and middle-aged population. Those who remain are predominantly elderly couples or seniors living alone. Previous studies have highlighted a link between loneliness, social isolation, and unhealthy dietary behaviors [36], such as low fruit and vegetable intake and poor diet quality. These living conditions may negatively impact the food consumption and overall nutrition of the elderly. Malnutrition negatively impacts the health of older adults, and dietary diversity represents a balanced intake of the six main food categories, as well as good and bad food quality, both of which are assessed as complementary. More than half of the participants had low DDS, suggesting that half of rural communities may be affected by a variety of factors and on-site eating habits and that they do not meet the Daily Dietary Guidelines and National Dietary Indicators [37]. This highlights the ongoing need for multiple strategies to increase the dietary balance and nutrition quality of rural populations.

### 4.1. Examination of Factors Affecting DDS

Previous research has explored the quality of meals in the CMS, emphasizing that balanced and adequate nutritional intake in older adults is a key factor for health. The HPA [38] has continued to establish “community nutrition promotion centers and sub-centers” in the county cities, seeking funding and expanding the number of community nutritionists, while integrating the community meal centers. These initiatives aim to provide sustainable nutrition care, supportive meal environments, and demonstration benefits, thereby enhancing the health and nutritional status of older adults. The combination of subsidies from the government and health resources from the healthcare system should gradually improve the low dietary diversity of rural communities. More than half (50.9%) of the population has a low dietary diversity, and previous studies have shown that there is a link between nutritional diversity and obesity [39], where persons who were overweight or obese accounted for 53.9% of the community’s population. Moreover, dietary diversity represents the quality of food intake and dietary balance; therefore, it is recommended that community residents prioritize a strategy that ensures a balanced intake of the six main food categories to meet the recommended intake of multiple micronutrients [40,41].

In rural communities with low dietary diversity, particular attention should be given to the nutritional status of older adults who are slower in walking, poor in grip, and poor in performing daily life functions. Previous studies have indicated that DDS is affected by factors such as age, gender, chewing ability, intellectual activity, living arrangements, economic conditions, and food availability [42,43,44]. Other factors such as grip strength, walking speed, musculoskeletal disorders, and daily life function assessment (BADL) all affect dietary diversity. In the rural context of Yunnan Prefecture, which is Taiwan’s most important agricultural production county [45], where up to 62% of the total cultivated land and 49.7% of the population is engaged in farming, it is possible that labor participation influences dietary patterns. The grip and walking tests in this study are both a condition for diagnosing weakness or musculoskeletal depression and are easy-to-obtain data, particularly the grip test. Previous studies indicated that lower grip strength in older adults was linked to other physical constraints, such as standing from chairs, walking, climbing stairs, and going out [46]. More scholars have indicated that grip is an essential biological marker for older adults [47] and has higher completion rates compared to walking tests [48]. Therefore, the grip is a detection indicator worth promoting in communities. The association of food diversity with calorie and protein intake is reflected in the past.

For DDS, a balanced diet used to evaluate health is already a recognized evaluation indicator and plays an important role in managing chronic diseases and preventing complications [49]. This study analyzed the relationship between rural population dietary diversity and the CCI, which, although not statistically significant (*p* = 0.056), showed a tendency to have a higher prevalence of low dietary diversity in the population. Additionally, dietary diversity is associated with better physical function performance [50], reduced dementia risk [51], and a decreased risk of depression through nutrients such as protein, vitamin D, calcium, folic acid, and antioxidants [52,53]. Furthermore, higher dietary protein intake is positively correlated with muscle mass [54,55]. This study shows that low DDS is associated with a higher prevalence of anemia, demonstrating that the risk of developing anemia can be improved by incorporating a diversified diet with adequate protein intake [51].

### 4.2. Relationship Between Malnutrition and Dietary Diversity

This study found that the rate of malnutrition among community residents was only 5.5%, consistent with the results of previous studies [56,57]. Former scholars used MNA-SF to evaluate the nutritional status of older adults in the community and conducted a plasma protein assessment, which showed that 21% of the senior community had a risk of malnutrition, with a prevalence rate of 3.4%, and, if a biochemical parameter was determined, an estimated prevalence of 3.5% [28]. According to the recommendations of previous studies [58,59], 26.4% (3.5% and 22.9%) of the community population were undernourished when diagnosed based on BMI. In older populations living in relatively healthy communities with increasing independence, the risk of malnutrition with MNA-SF screening is gradually reduced [60]. According to our research, 30% of the population has deliberately reduced their intake to ensure weight control (overweight to obese 53.9%). Moreover, the CMS has increased the nutrition of people who struggle with inadequate intake, underscoring the importance of community-wide meal services for sustainable nutrition care in older populations.

### 4.3. Policy and Future Intervention Implications

In response to the rapid increase in the aging population, municipal governments in each county have provided benefits to rural community older adults through long-term care policies. However, in urban or remote CMS centers, it has been found that remote communities are unable to submit plans for long-time care resources and subsidies due to challenges of equipment, catering personnel, site managers, or transportation. Thus, remote communities have fewer resources and may only provide a single main dish—often just plain noodles with minimal ingredients like a few carrots and a small portion of corn donated by residents—which is a meal but not a balanced diet. Around one-third to one-fourth of the elderly divide their lunch from the CMS center between lunch and dinner without supplementing with snacks. Previous studies have shown that factors affecting the participation of older adults in their meals are peers, distance from community centers [61], personality, and type of food offered, while community centers are influenced by resources and funding [62]. Given these challenges, it is crucial for the HPA of the MOHW, 2024, to pursue the community-based catering program under the “Community Nutrition Promotion Centers and Sub-Centers” scheme, with 124 community nutritionists involved by 113 years, to include a location-wide catering follow-up for this situation, and to provide additional support to vulnerable rural communities.

In our study, gender and living status (whether to live alone) (Table 2) did not affect the DDS. This may be due to differences in social background, living culture, and social support. Given that the local area is primarily agricultural, over half of the older adults in the community can farm and provide for themselves. The establishment of community care centers has also contributed significantly by serving as distribution hubs for the government’s most basic level of care for this population. In addition to addressing the meal needs of older adults, this initiative leverages “shared meals” to enhance opportunities for peer interaction among seniors [63]. It also promotes holistic health care, supports food and nutritional security, and ultimately improves the health and quality of life of older adults [64,65], facilitating aging in place.

Our findings also underscore the diet variability between urban and rural areas, especially among older adults residing in more remote areas. Limited access to varied food sources results in repetitive meals (largely based on seasonal produce) and reduced dietary diversity, leading to nutrient deficiencies. The interaction and complementarity of nutrients in a balanced diet play an important role in preserving physical function [51].

### 4.4. Limitations

This study is observational and fails to establish a causal relationship between dietary diversity and related factors, and will thus be subjected to further research in the future. In addition, this study recruited participants from community activity centers, resulting in a higher proportion of women than men. Consequently, the findings may have limited generalizability to the male population. Furthermore, this study used the 24HR questionnaire to verify food diversity, which may not fully represent daily eating habits, as the lifestyle and routines of older adults are very consistent and unchanging. Furthermore, the food frequency questionnaire was also utilized to strengthen the accuracy of dietary habit assessments. While over 40.0% of rural residents were reported to have inadequate caloric and protein intake, this figure may not fully reflect actual dietary insufficiencies. Finally, although we assessed socioeconomic status, certain critical factors that influence nutritional intake, such as social support networks, food delivery services, and seasonal food availability, were not directly measured in this study. However, we did evaluate variables such as living status, education, and income, which may serve as partial indicators of social support for older adults. Additionally, the impact of seasonal food availability was likely minimal in our study, as data collection occurred over a 6-month period, thereby reducing the potential influence of seasonal variations on nutritional intake. With regard to food delivery services, we acknowledge that access to these services may be influenced by proximity to urban centers, which could affect availability for participants.

### 4.5. Strengths

Our study has several strengths. First, it includes a large sample size compared to other community studies, enhancing the reliability of our findings. Additionally, the sample population is diverse, with data collected from multiple rural, suburban, and urban areas, ensuring a broader representation of different populations. Furthermore, the data collection is comprehensive, allowing for the simultaneous exploration of various factors related to DDS. Most notably, this study is the first to examine the relationship between distance and DDS, providing new insights into this association. Finally, a key aim of this study is to quantify the impact of distance, identifying 8 km as a critical determinant. These findings can help policymakers design more targeted and precise interventions by recognizing distance as a crucial factor in predicting malnutrition. Specifically, our results suggest the need to allocate more resources to rural communities located beyond the 8 km threshold, ensuring better accessibility and support for at-risk populations.

## 5. Conclusions

Distance from the community to the city center significantly influences the DDS of the residents in rural communities. Nearly half of the rural population has a low DDS. The implementation of this research project aimed to determine the feasibility of a daily nutrition evaluation and optimization program for the elderly in rural areas is crucial in the future.

## Figures and Tables

**Figure 1 nutrients-17-01318-f001:**
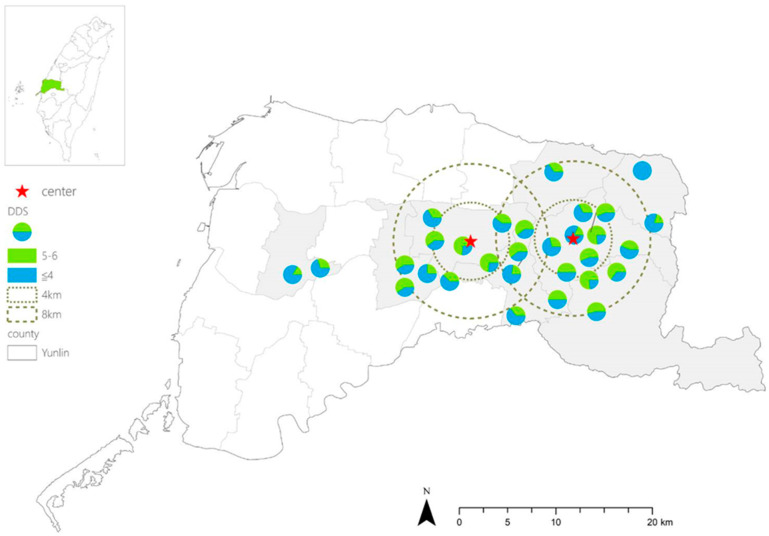
The circles are the locations of 30 communities, and the two red stars are the city centers (the Huwei Town Farmers Association and Douliu Railway Station). The color of the circles represents the DDS score.

**Table 1 nutrients-17-01318-t001:** Baseline characteristics of 567 community-dwelling adults.

	Total	Men	Women	*p*-Value
n	567	106 (18.7%)	461 (81.3%)	
Age (yr), mean (SD)	74.8 (8.7)	75.2 (8.7)	74.7 (8.7)	0.628
BMI, mean (SD)	24.7 (3.7)	24.8 (3.4)	24.7(3.8)	0.845
Living with, n (%)				
Alone	116 (20.5)	13 (12.3)	103 (22.3)	<0.001
Spouse	140 (24.7)	44 (41.5)	96 (20.8)	
Children	192 (33.9)	18 (17.0)	174 (37.7)	
Spouse and child	100 (17.6)	28 (26.4)	72 (15.6)	
Other	19 (3.4)	3 (2.8)	16 (3.5)	
Living distance to the city center, n (%)				
0–4 miles	154 (27.2)	31 (29.3)	123 (26.7)	0.1914
4–8 miles	255 (44.9)	53 (50.0)	202 (43.8)	
>8 miles	158 (27.9)	22 (20.7)	136 (29.5)	
Education Level, n (%)				<0.001
Illiterate	157 (27.7)	5 (4.7)	152 (33.0)	
Elementary school	212 (37.4)	44 (41.5)	168 (36.4)	
Junior high school	68 (12.0)	16 (15.1)	52 (11.3)	
Senior high school	85 (15.0)	20 (18.9)	65 (14.1)	
College or above	45 (8.0)	21 (19.8)	24 (5.2)	
Income per month, n (%)				0.074
No income	88 (15.5)	10 (9.4)	78 (16.9)	
≤300 USD	295 (52.0)	54 (50.9)	241 (52.3)	
>300 USD	184 (32.5)	42 (39.6)	142 (30.8)	
BADL score, mean (SD)	98.64 (4.9)	99.3 (3.2)	98.5 (5.1)	0.126
Independent, n (%)	508 (89.6)	99 (93.4)	409 (88.7)	0.316
Mild disability, n (%)	19 (3.3)	3 (2.8)	16 (3.5)	
Moderate disability, n (%)	40 (7.1)	4 (3.8)	36 (7.8)	
CCI, n (%)				
1	19 (3.3)	2 (1.9)	17 (3.7)	0.636
2	55 (9.7)	11 (10.4)	44 (9.5)	
≥3	493 (87.0)	93 (87.7)	400 (86.8)	
Sarcopenia, n (%)	118 (20.8)	23 (21.7)	95 (20.6)	0.803
Frailty, n (%)				0.086
Robust	353 (62.2)	72 (67.9)	281 (61.0)	
Pre-frailty	154 (27.2)	29 (27.4)	125 (27.1)	
Frail	60 (10.6)	5 (4.7)	55 (11.9)	
MNA-SF score	13.6 (SD 1.03)	13.6 (SD 0.9)	13.6 (SD 1.1)	0.348
Calorie intake (kcal/day)	1287 (SD 395)	1547 (SD 477)	1229 (SD 349)	<0.001
Protein intake (g/day)	49 (SD 19.2)	58.8 (SD 22.3)	46.8 (SD 17.8)	0.009
Fat intake (g/day)	47.5 (SD 20.8)	56.5 (SD 24.8)	45.5 (SD 19.2)	0.007
Carbohydrate intake (g/day)	159 (SD 53.3)	193.6 (SD 62.3)	151.2 (SD 47.7)	0.004
Dietary Diversity Score (DDS)	4.5 (SD 0.9)	4.7 (SD 0.8)	4.5 (SD 0.9)	0.048

BADL: Basic Activities of Daily Living, CCI: Charlson Comorbidity Index, MNA-SF: Mini Nutritional Assessment-Short Form.

**Table 2 nutrients-17-01318-t002:** Baseline characteristics and nutrition-related index according to low or high DDS.

	DDS ≤ 4		DDS > 4		*p*-Value
n	274		264		
Age, years; mean (SD)	75.66	(9.0)	73.88	(8.1)	0.016
≤75, n (%)	123	(44.9)	145	(54.9)	0.020
>75	151	(55.1)	119	(45.1)	
Female, n (%)	231	(84.3)	208	(78.8)	0.098
BMI, mean (SD)	24.9	(3.85)	24.6	(3.38)	0.399
Education Level, n (%)					
Illiterate	88	(32.1)	61	(23.1)	<0.001
Elementary school	116	(42.3)	83	(31.4)	
Junior high school	27	(9.9)	37	(14.0)	
Senior high school	29	(10.6)	53	(20.1)	
College or above	14	(5.1)	30	(11.4)	
Living Status, n (%)					
Alone	59	(21.5)	53	(20.1)	0.677
Not living alone	215	(78.5)	211	(79.9)	
Living distance to the city center, n (%)					
0–4 miles	70	(25.6)	79	(29.9)	<0.001
4–8 miles	113	(41.2)	136	(51.5)	
>8 miles	91	(33.2)	49	(18.6)	
Monthly income, n (%)					
≤300 USD	200	(73.0)	162	(61.4)	0.004
>300 USD	74	(27.0)	102	(38.6)	
BADL_score	98.2	(5.6)	99.1	(4.2)	0.031
Independent	246	(89.8)	253	(95.8)	0.022
Mild disability	25	(9.1)	9	(3.4)	
Moderate disability	3	(1.1)	2	(0.8)	
Charlson Comorbidity Index, n (%)					
1	10	(3.6)	8	(3.0)	0.056
2	17	(6.2)	32	(12.1)	
≥3	247	(90.1)	224	(84.8)	
Sarcopenia, n (%)					
No	203	(74.1)	227	(86.0)	0.001
Yes	71	(25.9)	37	(14.0)	
Calf circumference (cm)	32.7	(3.4)	33.1	(3.3)	0.113
6M Gait speed (s)	6.8	(3.7)	6.0	(3.6)	0.011
Hand grip strength (kg)	19.8	(6.9)	21.7	(7.2)	0.002
Frailty, n (%)					
Robust	159	(58.0)	179	(67.8)	0.031
Pre-frailty	79	(28.8)	65	(24.6)	
Frail	36	(13.2)	20	(7.6)	
MNA-SF score, mean (SD)	13.6	(1.0)	13.6	(1.0)	0.348
Well-nourished, n (%)	260	(94.9)	249	(94.3)	0.769
At risk of malnutrition, n (%)	14	(5.1)	15	(5.7)	
Dietary intake, mean (SD)					
Total calorie (kcal/day)	1116	(364)	1465	(345)	<0.001
Protein intake (g/day)	42.5	(18.6)	55.7	(17.6)	<0.001
Fat intake (g/day)	41.3	(20.3)	54.0	(19.2)	<0.001
Carbohydrate intake (g/day)	137.3	(47.8)	181.5	(49.3)	<0.001

DDS: Dietary Diversity Score. MNA-SF: Mini Nutritional Assessment-Short Form. BADL: Basic Activities of Daily Living.

**Table 3 nutrients-17-01318-t003:** Multivariate logistic regressions were used to evaluate the associations between the characteristics and risk of DDS.

	Multivariate Regression			Stepwise Selection		
	OR	95% CI	*p*-Value	OR	95% CI	*p*-Value
Female	1.08	0.62–1.87	0.798	-	-	-
Age over 75 y/o	1.08	0.65–1.77	0.770	-	-	-
BMI	1.01	0.95–1.08	0.655			
Education (compared to illiterate)						
Junior high school	1.23	0.73–2.09	0.074	-	-	-
Senior high school	0.72	0.36–1.45	0.136	-	-	-
living alone	0.96	0.58–1.59	0.859	-	-	-
Income ≥ USD 300/month	1.04	0.63–1.72	0.889	-	-	-
ADL mild disability	1.28	0.43–3.80	0.864	-	-	-
ADL moderate disability	1.99	0.81–4.87	0.275	-	-	-
CCI = 2 vs. 1	0.46	0.12–1.73	0.418	-	-	-
CCI ≥ 3 vs. 1	0.45	0.14–1.48	0.282	-	-	-
Sarcopenia	2.17	1.24–3.78	0.006	2.31	1.40–3.80	0.001
Pre-frailty	0.99	0.61–1.61	0.969	-	-	-
Frailty	0.96	0.44–2.12	0.928	-	-	-
At malnutrition risk	0.53	0.20–1.37	0.188	-	-	-
Distance to city center						
4–8 miles	0.53	0.31–0.89	<0.001	0.63	0.39–1.01	<0.001
>8 miles	1.80	1.02–3.18	<0.001	2.17	1.28–3.69	<0.001
CAL over 80% need	0.12	0.07–0.21	<0.001	0.09	0.06–0.15	<0.001

The associations were presented as adjusted odds ratios (aORs) and 95% confidence intervals (CIs). The *p* values were two sided, with *p* < 0.05 considered statistically significant. Multivariate logistic regression with stepwise selection was adopted to evaluate factors significantly associated with the risk of low DDS.

## Data Availability

The raw data supporting the conclusions of this article will be made available by C.-Y.C., S.-H.F., and C.-Y.W. on request.

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
