# Peer review of "Distance from the Community to the City Center Is a Vital Determinant of Dietary Diversity Score for Rural Community-Dwelling Older Adults in Taiwan"

_nutrients, 2025, doi:10.3390/nu17081318_

Round 1
Reviewer 1 Report
Comments and Suggestions for Authors
The study provides an interesting observation about the nutritional status determined by distance to city center. The article is written in a comprehensive fashion and the conclusions appear to be based on the results.
Some revisions might be considered:
Title: add the study location
Introduction: the introduction could also be improved by providing the localities of the studies. Clearly, there might be considerable variance in this issue worldwide, e.g. is rural England comparable to rural China?
Methods: the study appears biased towards women. Why was there no sex balancing included?
Line 172 and throughout: I would avoid providing decimals for nutrient intake from questionnaires. The data are certainly not accurate to this level?
Reviewer 2 Report
Comments and Suggestions for Authors
The study investigates an important public health issue, exploring how geographical factors influence the dietary diversity of older adults. The use of the Dietary Diversity Score (DDS) as a nutritional assessment tool is appropriate and aligns with established methods. The sample size (n = 567) is sufficient to provide meaningful insights into dietary patterns in rural settings.
Suggestions
- The study’s objectives could be more clearly defined—specifically, whether the focus is on nutritional interventions, policy implications, or community health strategies.
- While factors like sarcopenia and socioeconomic status are addressed, other potential confounders such as social support networks, food delivery services, or seasonal food availability are overlooked.
- While the sample size is adequate, the absence of a power calculation limits the ability to assess whether the study is sufficiently powered for subgroup analyses.
- The cutoffs for distance categories (0–4 km, 4–8 km, >8 km) need theoretical justification.
- While adjusted odds ratios are presented, effect sizes and clinical relevance are underexplored.
- The manuscript highlights reduced dairy and fruit consumption but lacks deeper insights into potential cultural or behavioral drivers for these trends.
- Streamline the conclusion to emphasize key findings, practical implications, and future research directions.
- While the findings are impactful, additional insights on how these results can inform policy or intervention strategies are lacking.
Reviewer 3 Report
Comments and Suggestions for Authors
Thank you for the article, it is interesting . But you will have to make some improvements. At row 36-37 you say, in the same phrase, that there are many overweight, obese and...energy deficient>? How did they get obese if they are energy deficient? Please, underline clearly in the abstract and introduction that the study was carried out in rural Taiwan. There are many differences from all points of view linked with geography, even though in the end we are all human beings. You localize the setting of the study a little bit to late in the flow of the article. At row 208 you say that rural dwellers are dairy deficient. Are milk products traditionally consumed in Taiwan? (I wonder). While economic and geographic factors are analyzed, you could delve deeper into how cultural dietary habits, food preferences, and social structures influence dietary diversity among rural older adults. This would provide a more holistic understanding of the issue.
Than, from row 191 you describe factors predicting DDS. Interesting, though I thought that more interesting would be to see what is predicting a lower DDS. If the population has a low DDS, than what?! I know there are studies, but what did you find here?
Another suggestion for improvement is to underline that the study remains observational. You should emphasize the limitations of causality more explicitly and, if possible, suggest further longitudinal or intervention-based studies to strengthen the findings.
Round 2
Reviewer 3 Report
Comments and Suggestions for Authors
Thank you for answering to all our suggestions, the article has improved.